# Clinical Outcomes of HER2-Negative Metastatic Breast Cancer Patients in Italy in the Last Decade: Results of the GIM 13-AMBRA Study

**DOI:** 10.3390/cancers16010117

**Published:** 2023-12-25

**Authors:** Marina Elena Cazzaniga, Paolo Pronzato, Domenico Amoroso, Antonio Bernardo, Laura Biganzoli, Giancarlo Bisagni, Livio Blasi, Emilio Bria, Francesco Cognetti, Lucio Crinò, Michelino De Laurentiis, Lucia Del Mastro, Sabino De Placido, Alessandra Beano, Francesco Ferraù, Silva Foladore, Rosachiara Forcignanò, Teresa Gamucci, Ornella Garrone, Alessandra Gennari, Monica Giordano, Francesco Giotta, Filippo Giovanardi, Luciano Latini, Lorenzo Livi, Paolo Marchetti, Rodolfo Mattioli, Andrea Michelotti, Filippo Montemurro, Carlo Putzu, Ferdinando Riccardi, Giuseppina Ricciardi, Emanuela Romagnoli, Giuseppina Sarobba, Simon Spazzapan, Pierosandro Tagliaferri, Nicola Tinari, Giuseppe Tonini, Anna Turletti, Claudio Verusio, Alberto Zambelli, Giorgio Mustacchi

**Affiliations:** 1Fondazione IRCCS San Gerardo dei Tintori, 20900 Monza, Italy; 2School of Medicine and Surgery, Università Milano Bicocca, 20900 Monza, Italy; 3Oncologia Medica IRCCS IRST San Martino, 16100 Genova, Italy; paolo.pronzato@hsanmartino.it (P.P.); lucia.delmastro@hsanmartino.it (L.D.M.); 4Oncologia Medica, Ospedale Versilia USL Nord Ovest Toscana, 55041 Lido di Camaiore, Italy; domenico.amoroso@uslnordovest.toscana.it; 5Oncologia Istituti Clinici Scientifici Maugeri Spa Società Benefit, 27100 Pavia, Italy; antonio.bernardo@fsm.it; 6U.O. Oncologia Medica, Ospedale Santo Stefano, 59100 Prato, Italy; lbiganzoli@usl4.toscana.it; 7Oncologia Medica, IRCCS Arcispedale S. Maria Nuova, 42121 Reggio Emilia, Italy; bisagni.giancarlo@asmn.re.it; 8Oncologia Medica, ARNAS Civico–Di Cristina-Benfratelli, Presidio Ospedaliero ‘Civico e Benfratelli’, 20121 Palermo, Italy; livioblasi@tiscali.it; 9Oncologia Medica, A.O.U. Integrata Verona, Ospedale Borgo Roma, 37100 Verona, Italy; emiliobria@yahoo.it; 10Oncologia Medica 1, Istituto Nazionale Tumori “Regina Elena”, 00042 Roma, Italy; 11Oncologia, Ospedale S. Maria della Misericordia, 06121 Perugia, Italy; lucio.crino@ospedale.perugia.it; 12Breast Oncology Unit, INT Fondazione Giovanni Pascale-IRCCS, 80013 Napoli, Italy; m.delaurentiis@breastunit.org; 13Oncologia Medica, Dipartimento di Medicina Clinica e Chirurgia, Università degli Studi Federico II, 80013 Napoli, Italy; deplacid@unina.it; 14Oncologia Medica, Città della Salute e della Scienza, ASL “Città di Torino”, 10024 Torino, Italy; alessandra.beano@gmail.com; 15Oncologia Medica, Ospedale S. Vincenzo, 98039 Taormina, Italy; ferrau@oncologiataormina.it; 16SSD Oncologica e dell’apparato Riproduttivo Femminile, Azienda Sanitaria Universitaria “Giuliano Isontina, 34121 Trieste, Italy; silva.foladore@aas1.sanita.fvg.it; 17Oncologia Medica, Ospedale Vito Fazzi, 73100 Lecce, Italy; rosachiaraforcignano@gmail.com; 18Oncologia Medica, ASL Roma 2, 00042 Roma, Italy; t.gamucci@libero.it; 19Oncologia Medica, Fondazione IRCCS Cà Granda Ospedale Maggiore Policlinico, 20100 Milano, Italy; ornella.garrone@policlinico.mi.it; 20Oncologia Medica, Azienda Ospedaliero-Universitaria Maggiore della Carità di Novara, Dipartimento di Medicina e Chirurgia, Università degli Studi del Piemonte Orientale, 28100 Novara, Italy; alessandra.gennari@uniupo.it; 21SC Oncologia, Asst-Lariana, 22100 Como, Italy; monica.giordano@asst-lariana.it; 22Oncologia, IRCCS Istituto Tumori ‘Giovanni Paolo II’, 70100 Bari, Italy; f.giotta@oncologico.bari.it; 23UOS Day Hospital Oncologico, AUSL Reggio Emilia, Presidio Ospedaliero di Guastalla, 42016 Guastalla, Italy; filippo.giovanardi@ausl.re.it; 24Oncologia, Ospedale di Macerata, 62100 Macerata, Italy; 25Dipartimento di Scienze Biomediche, Sperimentali e Cliniche ‘Mario Serio’, Università degli Studi di Firenze, UOC Radioterapia, A.O.U. Careggi, 50100 Firenze, Italy; lorenzo.livi@unifi.it; 26Oncologia Medica, IDI IRCCS, 00042 Roma, Italy; paolo.marchetti@ospedalesantandrea.it; 27Oncologia Medica, A.O. Ospedali Riuniti Marche Nord-Ospedale Santa Croce, 61032 Fano, Italy; 28Oncologia Medica, Azienda Ospedaliera Pisana, 56121 Pisa, Italy; a.michelotti@ao-pisa.toscana.it; 29Fondazione del Piemonte per l’Oncologia-Istituto di Ricovero e Cura a Carattere Scientifico (I.R.C.C.S.), 10024 Torino, Italy; filippo.montemurro@ircc.it; 30Oncologia Medica, AOU Sassari, 07100 Sassari, Italy; 31Oncologia Medica, Azienda Ospedaliera ‘A. Cardarelli’ (AORN), 80013 Napoli, Italy; 32Oncologia Medica, Ospedale Papardo, 98121 Messina, Italy; 33Oncologia Medica, AV Macerata, 62100 Macerata, Italy; emanuela.romagnoli@sanita-marche.it; 34Oncologia Medica, Ospedale ‘San Francesco’, Azienda Sanitaria Locale 3 Nuoro, 08100 Nuoro, Italy; 35Centro di Riferimento Oncologico, IRCCS, 33081 Aviano, Italy; spazzapan@cro.it; 36Oncologia Medica, A.O.U. ‘Mater Domini’, 88100 Catanzaro, Italy; tagliaferri@unicz.it; 37Oncologia Medica, Policlinico ‘SS. Annunziata’, 66100 Chieti, Italy; nicola.tinari@unich.it; 38Policlinico Universitario Campus Biomedico, 00042 Roma, Italy; g.tonini@unicampus.it; 39Presidio Ospedaliero Martini, ASL TO1, 10024 Torino, Italy; 40Oncologia Medica, ASST della Valle Olona, Ospedale di Saronno, 21047 Saronno, Italy; claudio.verusio@asst-valleolona.it; 41Oncologia Medica, ASST Papa Giovanni XXIII, Ospedale Papa Giovanni XXIII, 24100 Bergamo, Italy; alberto.zambelli@hunimed.eu; 42Università degli Studi di Trieste, 34100 Trieste, Italy; mail@mustacchioncology.it

**Keywords:** metastatic breast cancer, HER2 negative, progression free survival, time to treatment change, overall survival

## Abstract

**Simple Summary:**

Breast cancer is one of the most common oncological diseases among women in Western Countries and Italy as well. GIM 13-AMBRA is a patient journey study regarding how the prognosis of metastatic breast cancer patients has changed in the last decades as a result of the introduction of new drugs in clinical practice. This study also provides information regarding the different natural histories of breast cancer according to the presence or absence of hormone receptors and HER2 receptors.

**Abstract:**

GIM 13-AMBRA is a longitudinal cohort study aimed at describing therapeutic strategies and the relative outcome parameters in 939 HER2-ve MBC patients. Taxanes–based regimens, or taxanes + targeted agents, mainly Bevacizumab, were the preferred first choice in both Luminal (30.2%) and TNBC (33.3%) patients. The median PFS1 was 12.5 months (95% CI 16.79–19.64), without any significant difference according to subtypes, while the median Time to first Treatment Change (TTC1) was significantly lower in TNBC patients (7.7 months—95% CI 5.7–9.2) in comparison to Luminal A (13.2 months, 95% CI 11.7–15.1) and Luminal B patients (11.8 months, 95% CI 10.3–12.8). PFS2 was significantly shorter in TNBC patients (5.5 months, 95% CI 4.3–6.5 vs. Luminal A—9.4, 95% CI 8.1–10.7, and Luminal B—7.7 95% CI 6.8–8.2, F-Ratio 4.30, *p* = 0.014). TTC2 was significantly lower in patients with TNBC than in those with the other two subtypes. The median OS1 was 35.2 months (95% CI 30.8–37.4) for Luminal A patients, which was significantly higher than that for both Luminal B (28.9 months, 95% CI 26.2–31.2) and TNBC (18.5 months, 95% CI 16–20.1, F-ratio 7.44, *p* = 0.0006). The GIM 13—AMBRA study is one of the largest collections ever published in Italy and provides useful results in terms of time outcomes for first, second, and further lines of treatment in HER2- MBC patients.

## 1. Introduction

Breast cancer (BC) is currently the leading cause of cancer–related death among women worldwide [1]. Stage and Region distribution [2] highlights that, in Italy, incidence accounts for approximately 55,000 newly diagnosed cases per year, and mortality accounts for 13,000 deaths [3]. Despite significant progress in the treatment of the primary tumor, approximately 30% of breast cancers are destined to develop distant metastases [4,5]. The clinical course of metastatic breast cancer (MBC) is very heterogeneous in terms of the growth rate and response to systemic therapies, and treatment remains a palliative cure. The median survival time is approximately 2 years for some subtypes. There is no standard chemotherapy strategy for HER2- tumors, which represent 80–85% of all BCs cases [5]. The only available Italian ‘real life’ study is the IRIS study [6], which described the relapse patterns and treatment modalities in a longitudinal cohort of 539 metastatic patients enrolled between 1999 and 2001. Randomized clinical trials (RCTs) remain the cornerstone in determining the efficacy and safety of new treatments. However, they are not fully representative of MBC patients, especially in terms of the exclusion criteria. Outcome parameters usually adopted in RCTs, such as Progression Free Survival (PFS), are based on time points defined a priori, a situation that cannot be applied in observational studies. The GIM 13-AMBRA study is a longitudinal cohort study aiming to describe therapeutic choices for HER2- MBC in real life in Italy. The main objective of the GIM 13-AMBRA study was to describe the therapeutic strategies in terms of first, second, and subsequent lines of treatment in a cohort of HER2- MBC patients receiving at least one line of chemotherapy (CHT) and the related outcome measures, including Time to Treatment Change (TTC), which could be used in place of PFS in future real-world studies.

## 2. Materials and Methods

### 2.1. Study Design

The GIM 13 AMBRA is a longitudinal cohort study that collected data from the first 50 consecutive HER2-MBC patients who started a first, second, or subsequent line CHT between January 2012 and December 2016. A total of 42 centers were selected from the 192 national Centers listed in the “Libro Bianco 2012 of the Italian Association of Medical Oncology—AIOM)”, according to hospital type and geographical distribution. Eligible patients were females aged 18 years or older with newly diagnosed MBC who provided written informed consent. All centers were authorized by their Ethical Committees (ECs) after approval from the Coordinating Center EC (CE Brianza).

### 2.2. Objectives

The primary objective was to describe the strategies in terms of the first, second, and subsequent lines of treatment in patients receiving at least one chemotherapy line (CHT) and the relative outcome parameters. Secondary objectives were to (1) evaluate any possible correlations between the choice of treatment, both in the adjuvant phase and for metastatic disease and patient characteristics (age, menopausal status, and comorbidities); (2) estimate recurrence patterns and clinical outcomes (TTC; PFS and OS); and (3) evaluate adherence to literature recommendations for therapeutic sequences in the clinical practice. Here, we present the results of the primary objective.

### 2.3. Definitions

HER2 and HR status were derived from the pathological report of the primary tumor tissue or the metastatic one in the case of de novo disease. Tumor subtypes were defined according to the definitions provided by Prat et al. [7,8].

### 2.4. Statistical Analysis

The primary endpoint was the distribution of drugs and regimens, and qualitative data were described using frequency and percentage distribution. The clinical outcomes were Disease–free Survival (DFS), defined as the time between primary diagnosis and death; Progression–Free Survival at first–line (PFS1) or second–line treatment (PFS2), defined as the time between first/second–line therapy start and time to progression, as assessed by the investigator or censored at date of latest news; and Time to Treatment Change of first–line (TTC1) or second–line (TTC2) therapy, defined as the time between the start date, declared by Investigator, of first or second–line treatment and the date, not defined a priori due to the observational design of the study, of subsequent therapy start. OS was defined as the time between the date of diagnosis of metastatic disease and the date of death (any cause) or censored to the date of the latest news. All these variables will be implemented using Kaplan–Meyer survival curves, and when indicated, the differences will be evaluated using Fisher’s exact test. Analyses were performed using the NCSS^®^ 12 Statistical Software 2018 (Kaysville, UT, USA). Continuous variables were evaluated using descriptive statistics (including the number of patients, mean, standard deviation, median, minimum, 25th and 75th percentiles, and maximum). Categorical variables were evaluated using frequencies and percentages. The F-ratio, defined as the ratio of the between–group variance (MSB) to the within–group variance (MSW), was used to determine any differences between groups. Data, which include patient demographics and tumor characteristics, outcomes, and treatment strategies, were implemented on an electronic platform specifically set up on a dedicated website. Considering that approximately 15,000 new cases of MBC are treated in Italy every year, a sample of 1000 cases is representative of the entire Italian population. One year of observation will make it possible to achieve both the primary and secondary objectives.

## 3. Results

Between May 2015 and September 2020, 1071 patients were enrolled, of whom 132 (12.3%) were not considered eligible due to (1) incomplete information about the 1st-line setting and (2) other reasons. The median age at primary tumor diagnosis was 51.9 years (range: 50.6–52.9). Appendix A summarizes the characteristics of patients and tumors characteristics at diagnosis, according to breast cancer subtypes.

No difference in the median age or stage at diagnosis was observed among the 3 different subtypes. (Appendix A). Most patients received adjuvant CHT (71.8%), mainly a combination of anthracycline + taxanes (305, 31.5%) or anthracycline + other drugs (266, 28.3%). Interestingly, a significant percentage (261, 67.6%) of Luminal A patients received adjuvant chemotherapy, and the preferred choice was an anthracycline-based combination (125, 32.4%), whereas the preferred adjuvant choice for TNBC patients was an anthracycline-taxane regimen (305, 31.5%) (Table 1).

Median Disease–Free Survival (DFS) was 57.2 months (95% CI 53.2–63.8) and was significantly longer in Luminal A (87 months, 95% CI: 75.3–91.7) vs. Luminal B (50.7 months, 95% CI 46.4–56.5; HR = 0.71 (0.62–0.82)) and TNBC patients (24.3 months, 95% CI 21.6–29.2; TNBC/Luminal A: HR = 2.22 (1.74–2.83); TNBC/Luminal B: HR = 1.74 (1.39–2.17)) (Figure 1 and Table 2).

Information about the first site of relapse was available for 922 out of the 939 patients (98.2%). Luminal A and Luminal B patients presented a different pattern of relapse, with Luminal A patients relapsing mainly at the bone (31.9%), whereas Luminal B patients relapsed at both bone (24.0%) and bone + viscera (20.8%) as well as viscera alone (23.5%) (Table 3).

As first-line treatment, Luminal patients mainly received ET, alone (42%), or sequentially to CHT (29.2%). Taxanes-based regimens, or taxanes + targeted agents, mainly Bevacizumab, were the preferred first choices in both Luminal (30,2%) and TNBC (33.3%) patients (Table 4).

We also collected the main adverse events (AEs) for 1st-line treatments: Grade 3 toxicity (any type) was mainly present during the first cycle (11.0%) and decreased during the subsequent cycles, while the incidence of Grade 4 AEs was negligible (max 0.72% in patients receiving >6 cycles).

The median PFS1 was 12.5 months (95% CI 16.79–19.64), without any significant difference according to subtypes (Luminal A 18.03 months, Luminal B 15.95 months, and TNBC 14.04 months), while median TTC1 was significantly lower in TNBC patients (7.7 months—95% CI 5.7–9.2) compared to Luminal A (13.2 months, 95% CI 11.7–15.1) and Luminal B patients (11.8 months, 95% CI 10.3–12.8) (Figure 2).

As expected, second–line treatments were significantly different in Luminal A & B vs. TNBC patients: most of the Luminal patients received endocrine therapy (29%), or, in the case of CHT, Capecitabine ± Vinorelbine, whereas TNBC was mainly treated with platinum–based regimens (22.4%), or with Capecitabine ± Vinorelbine (21%). The PFS of second–line treatment (PFS2) was significantly shorter in TNBC patients (5.5 months, 95% CI 4.3–6.5 vs. Luminal A—9.4, 95% CI 8.1–10.7 and Luminal B—7.7 95% CI 6.8–8.2, F-Ratio 4.30, *p* = 0.014). Similarly, the Time to second treatment change (TTC2) was significantly lower in TNBC patients than in the other 2 subtypes (Figure 3 and Figure 4).

The median OS from primary diagnosis was 10 years for Luminal A patients and 7 and 3.6 years for Luminal B and TNBC patients, respectively (*p* = 0.0006). (Figure 5).

The median OS from the first progression (PD1) was 35.2 months (95% CI 30.8–37.4) for Luminal A patients, significantly higher than both Luminal B (28.9 months, 95% CI 26.2–31.2) and TNBC ones (18.5 months, 95% CI 16–20.1, F-ratio 7.44, *p* = 0.0006) (Figure 6).

Outcome parameters for DFS, TTC1, TTC2, OS, and OS from PD1 are summarized in Appendix A.

## 4. Discussion

The GIM 13-AMBRA study is a large-scale Italian real-world cohort study on HER2- MBC treatments and a unique opportunity for evaluating ‘historical’ PFS results under commonly used first, second, and subsequent lines of chemotherapy. Considering the study design and the main inclusion criteria, our population is representative of a high-risk cohort of BC patients, and this was confirmed by the high percentages of N + ve (61.76%) and Grade 3 (50.47%) tumors at diagnosis.

In the years under study, more than 70% of the patients were treated with CHT in the adjuvant setting, regardless of molecular subtype: a large use of CHT especially in Luminal A patients (67.6%) reflects the lack of tools, like genomic signatures, able to identify patients at a higher risk of relapse for whom CHT remains mandatory [7]. Anthracyclines represented the most commonly used agents in Luminal A patients, especially in combination with 5-Fluorouracil and Cyclophosphamide (44.2%), and recent data from the GIM 2 study [8] no longer support the use of this drug in adjuvant treatments considering the absence of improvement in outcome.

In terms of median PFS of first–line treatment (PFS1), our results are very similar to those reported in the ESME study for the Luminal patients, which showed a median PFS under first–line therapy of 9.6 months for the whole population (95% CI, 9.4–9.9) and 10.7 months (95% CI, 10.5–11.0) in the HR + /HER2- subgroup. The advent of Cyclin–Dependent Kinase 4/6 inhibitors (CDK 4/6i), Palbociclib, Abemaciclib and Ribociclib, in combination with Aromatase Inhibitors in endocrine-sensitive patients has dramatically changed the outcome of Luminal patients: median PFS1 is now around 25 months [9,10,11], thus far, also resulted in delaying the time to meet CHT needs. In addition, Ribociclib in combination with letrozole showed an improvement in OS (63.9, 95% CI 52.4 to 71.0) in the Monaleesa-2 trial [9]. Whether CDK 4/6i will definitely change the natural course of Luminal cancers is still a matter of debate: preliminary data from some small and heterogeneous trials [12] suggest that there is no recommended sequence at the category 1 level in international guidelines. Randomized clinical trials are still underway, and subsequent treatments will be used in patients with progressive disease under CDK 4/6i + ET. The approximately 7-month PFS obtained in the phase 2 ByLieve study, which evaluated the efficacy of alpelisib in patients who had previously received CDKi-based therapy, indicated that alpelisib + fulvestrant might be effective in PIK3CA mutant patients [13].

On the contrary, very little has changed for the TNBC patients over the last decade: median PFS1 was 8.8 months (95% CI 6.7–10.2) vs. 4.8 months (95% CI, 4.6–5.1) described in the ESME collection. No details regarding the type of CHT have been reported in this latter study; thus far, we cannot derive conclusive affirmations.

In our study, the median OS from primary diagnosis was 10 years for Luminal A patients and 7 and 3.6 years for Luminal B and TNBC patients, respectively (*p* = 0.0006). Since 2008, different drugs have demonstrated an OS benefit in MBC patients, particularly eribulin in HER2-VEve MBC [14]. In the ESME cohort, the median OS of the entire study population was 37.22 months (95% CI, 36.3–38.04) and 14.52 months (95% CI, 13.70–15.24) in the TNBC sub–cohort. The Authors highlighted the lack of significant improvement in luminal (HR+/HER2-VE) and TNBC triple-negative subtypes (HR-/HER2-VE) over the considered period (2008–2014), suggesting that these results may be linked to the too recent introduction of some innovative drugs, or to the limited impact of some others. Our results indicate the use of some targeted agents, mainly bevacizumab, in both Luminal (20%) and TNBC patients (25%) patients. Despite a lack of evidence of OS improvement in RCTs [15], various real-world studies showed different results: a cost–effectiveness analysis clearly indicated that the combination of bevacizumab plus paclitaxel was likely to be cost-effective compared with paclitaxel alone for the first–line treatment of HER2-VEve MBC [16] and the ATHENA trial reported a median TTP of 9.5 months (95% CI: 9.1–9.9) [17].

The identification of different molecular subtypes of TNBC by Lehmann et al. [18] allowed a different and more tailored approach according to the related gene expression in recent years, such as immunotherapy in PDL-1+ve patients. In the IMPassion130 trial [19], median PFS was longer in the atezolizumab–nab-paclitaxel group than in the placebo group (ITT: median 7.2 months vs. 5.5 months, HR 0.8; PD-L1+: 7.5 vs. 5.0 months, HR 0.62), even if the final OS analysis showed a statistically significant OS improvement only in the PD-L1+ population. Similar results have been obtained with pembrolizumab plus chemotherapy: median PFS was 9.7 months in the pembrolizumab + chemotherapy arm and 5.6 months in the placebo arm (HR 0.65), and median OS was 23.0 vs. 16.1 months (HR 0.73). In light of these results, compared to our data obtained at different times and with different drugs, we can conclude that there is still a long way to go for mTNBC patients.

One of the aims of our study was also to investigate the role of TTC as a surrogate for PFS in real–world studies. The median PFS1 and median TTC1 were very similar in all patients. We performed an analysis focusing on the comparison between TTC1 and PFS1 of Paclitaxel + Bevacizumab (PB) as an example of TTC performance in real–world studies (RWS): median TTC was 9.36 months (40.67 weeks) and median PSF 10.8 (46.92 weeks). The difference at 6.2 weeks was not significant (Wilcoxon rank–sum test, α 0.050, *p* = 0.089) and was within the preplanned boundary limit. PFS1 for the PB regimen was 1-month shorter vs. that reported in the ECOG 2100 trial [15] and seems reliable for the clinical setting due to the different populations enrolled. Based on these results, we suggest that TTC is a valid surrogate endpoint of PFS in RWS and should be considered in future observational trials.

GIM 13-13 AMBRA data provide the national and international oncology community with real-life survival data that can be used as strong references for future clinical trials and for a better understanding of the real improvement of new drugs compared to the control arms.

This study has some limitations that can be summarized as follows: (1) the retrospective nature of the trial, (2) information about subtype classification, (3) retrospective collection of treatment decisions and time points, (4) representativeness of all MBC patients across Italy, and (5) absence of patient–reported outcomes.

## 5. Conclusions

The GIM 13-AMBRA study is one of the largest collections ever published in Italy and has provided useful results in terms of time outcomes for first, second, and further lines of treatment in HER2-VEve MBC patients. It would be interesting to repeat this collection in a more recent time, especially to evaluate the impact of new drug introduction into clinical practice.

## Figures and Tables

**Figure 1 cancers-16-00117-f001:**
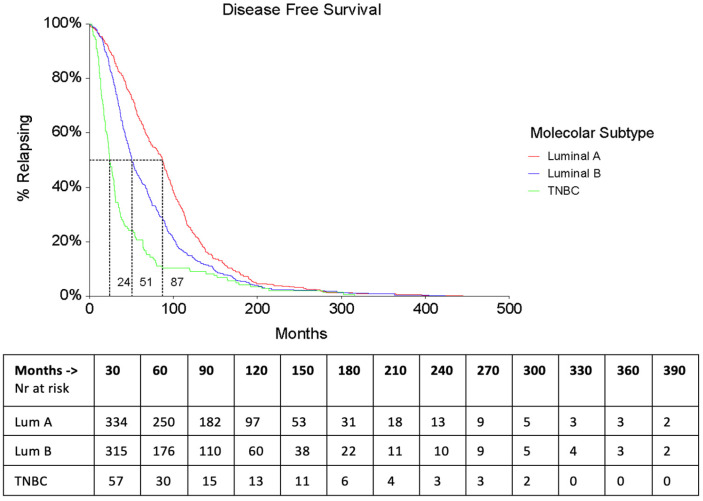
Disease-Free Survival according to subtypes.

**Figure 2 cancers-16-00117-f002:**
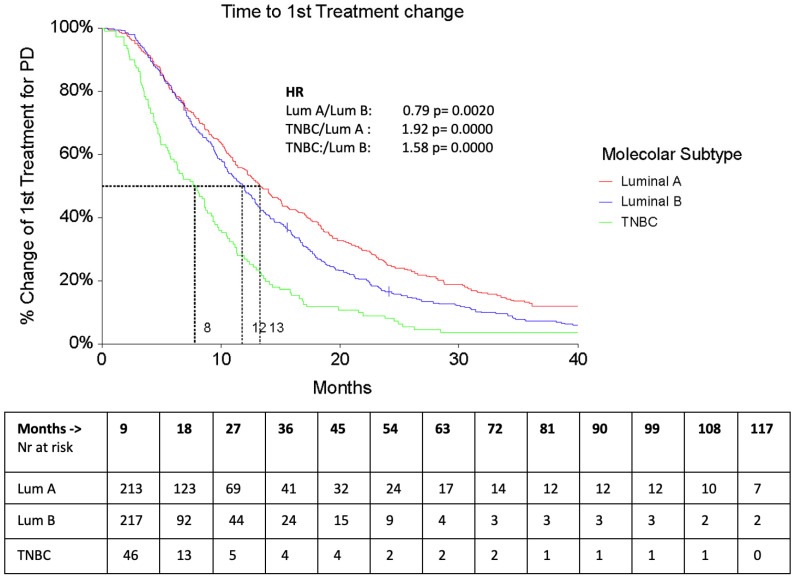
TTC 1 according to subtypes.

**Figure 3 cancers-16-00117-f003:**
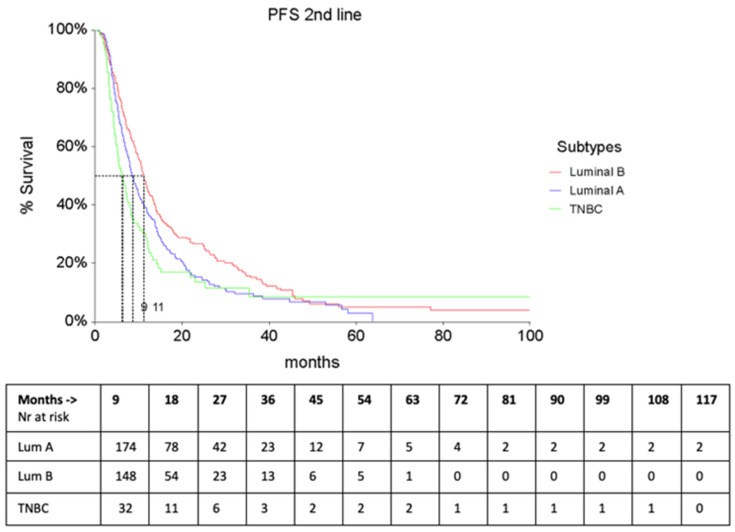
PFS2 according to subtypes.

**Figure 4 cancers-16-00117-f004:**
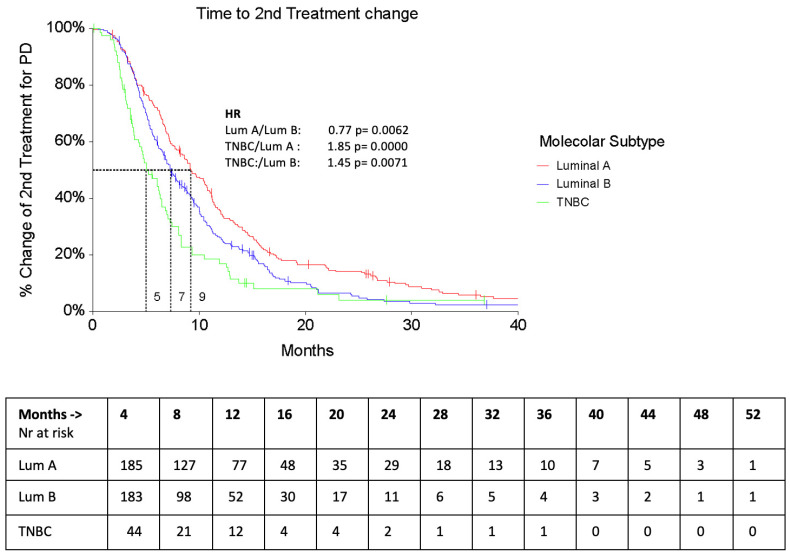
TTC2 according to subtypes.

**Figure 5 cancers-16-00117-f005:**
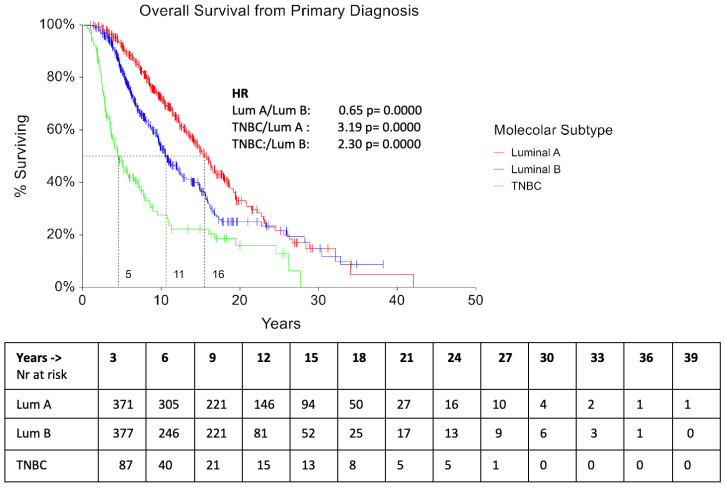
Overall Survival (OS) according to subtypes.

**Figure 6 cancers-16-00117-f006:**
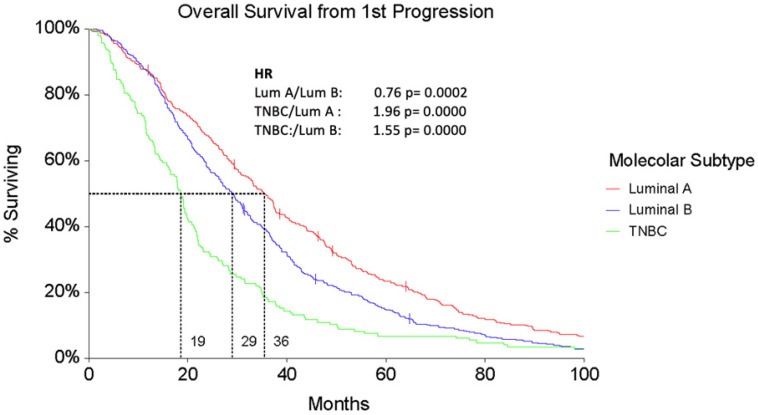
Overall Survival from first progression (OS1) according to subtypes.

**Table 1 cancers-16-00117-t001:** Adjuvant therapies in the whole population and according to subtypes.

	Luminal A (*N* = 386)	Luminal B *N* = 408	TNBC *N* = 145	All Patients *N* = 939
None	13 (3.4%)	13 (3.2%)	14 (9.6%)	939
Endocrine therapy alone	112 (29.0%)	91 (22.3%)	0	203 (20.7%)
CHT + ET	245 (63.5%)	279 (68.4%)	4 ^1^	528 (53.9%)
**CHT regimens**				
Anthracycline-based	125 (32.4%)	109 (26.7%)	32 (22.1%)	266 (28.3%)
Anthracycline + Taxanes	93 (24.1%)	137 (33.6%)	75 (51.7%)	305 (31.5%)
taxanes ^2^	4 (1.0%)	16 (3.9%)	7 (4.8%)	27 (2.9%)
CMF	38 (9.8%)	40 (9.8%)	16 (11.0%)	94 (10.0%)
Others	1	2	1	4

^1^ A total of 4 patients with very low ER and PgR expression (<5%) received ET. ^2^ taxanes alone or in combination with other drugs are different from anthracyclines.

**Table 2 cancers-16-00117-t002:** Hazard Ratiologrank test for DFS.

Subtypes Comparison	Cox Mantel HR (95% CI)	Cox Mantel Logrank Test Chi^2^
Luminal A/Luminal B	0.71 (0.62–0.82)	23.15
TNBC/Luminal A	2.22 (1.74–2.83)	70.59
TNBC/Luminal B	1.74 (1.39–2.17)	33.62

**Table 3 cancers-16-00117-t003:** Patterns of relapse according to subtypes.

	Luminal A (*N* = 386)	Luminal B *N* = 408	TNBC *N* = 145	All Patients *N* = 939
Bone	123 (31.9%)	98 (24.0%)	9 (6.2%)	230 (30.4%)
Bone + soft tissue	21 (5.4%)	22 (5.4%)	7 (4.8%)	50 (6.6%)
Viscera	84 (21.8%)	96 (23.5%)	39 (26.9%)	219 (28.9%)
Viscera + soft tissue	29 (7.5%)	33 (8.1%)	23 (15.9%)	85 (11.2%)
Viscera + bone	68 (17.6%)	85 (20.8%)	13 (8.9%)	166 (17.7%)
Soft tissue	53 (13.7%)	62 (15.2%)	46 (31.7%)	161 (21.3%)
Other	4 (1.0%)	3 (0.7%)	1 (0.6%)	4 (0.5%)
Not specified				17

**Table 4 cancers-16-00117-t004:** First–line treatment choices according to subtypes.

	Luminal A & B *N* = 460 (%)	TNBC *N* = 141 (%)
ET alone	334 (42%)	0
Chemo + ET	232 (29.2%)	0
**Chemotherapy regimens**		
Capecitabine +/− Vinorelbine	82 (17.8)	24 (17.02)
Paclitaxel + Bevacizumab	139 (30.2)	47 (33.33)
Platinum-based	23 (5)	25 (17.7)
Anthracycline-based	44 (9.5)	5 (3.5)
Anthracycline + Taxanes	19 (4.13)	2 (1.4)
Taxanes	133 (28.9)	22 (15.6)
CMF	4 (0.8)	7 (4.9)
Other	16 (3.4)	9 (6.3)

## Data Availability

Data supporting reported results can be found at Oncotech.

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
