# Peer review of "Clinical Outcomes of HER2-Negative Metastatic Breast Cancer Patients in Italy in the Last Decade: Results of the GIM 13-AMBRA Study"

_cancers, 2023, doi:10.3390/cancers16010117_

Round 1

Reviewer 1 Report

Comments and Suggestions for Authors

1- Check the name of the study in the title and text and standardize the writing. In each place it is written in a different way (GIM-13 - AMBRA? GIM 13 AMBRA? GIM 13 – AMBRA?). The same for HER2-ve or HER2-.

2- Summary: Although the meaning of the acronyms PFS, OS and TNBC are known worldwide and can be omitted, the acronym TTC should be described when first used. If it is necessary to reduce the size of the text, remove the “F-ratio” description.

3- Introduction: objective and informative. No modifications suggested.

4- Methods: why did they restrict only the first 50 patients? Were there hospitals with more than 50 cases? Others with less? Eventually, add this information to the supplementary document.

Line 78: correct typo (space after parenthesis): “characteristics ( age, menopausal status, comorbidities);”

If the number of patients screened was 1,071, why do they mention 1,500 in the text? In my opinion, 1,071 remains a significant sample.

Describe what the F-ratio is about, which will be mentioned later. How to interpret the results?

5- Results: The types of chemotherapy used were very well described. However, the description of ET was superficial. How many patients used tamoxifen or aromatase inhibitors? How many combined fulvestrant? How many used CDK4/6 inhibitors?

6- Discussion: in the third paragraph, although data using iCDK4/6 are mentioned, there was a lack of a comment linking the study carried out and data from the literature. Did the authors evaluate the use of these medications? Or were they not yet available at the time of the study? What impact does this information have on current practice?

Given the clinical benefits of using T-DXd in patients with HER2-low tumors, a description of this population and possibly a sub-analysis would be interesting. If it is the result of a future study, comment about it in the Discussion.

In line 238, in the paragraph on immunotherapy, note that the use of these medications is based on the expression of PD-L1 in the immunohistochemical study, which is not mentioned in the text (just “gene expression”). I suggest adding this information to the text.

Conclusions: objective and consistent with the study. No modifications suggested.

Author Response

1- Check the name of the study in the title and text and standardize the writing. In each place it is written in a different way (GIM-13 - AMBRA? GIM 13 AMBRA? GIM 13 – AMBRA?). The same for HER2-ve or HER2-. Thank you, we have modified all the acronims in HER2-, with the first explanation in the title

2- Summary: Although the meaning of the acronyms PFS, OS and TNBC are known worldwide and can be omitted, the acronym TTC should be described when first used. If it is necessary to reduce the size of the text, remove the “F-ratio” description. Thank you, we put the descrition of TTC1 in the abstract, even if it was detailed in the text. Same for TTC2

3- Introduction: objective and informative. No modifications suggested.

4- Methods: why did they restrict only the first 50 patients? We limited to the first 50 patients because, taken as a whole and considering the high numbers of Centers involved into this study, we believed that 50 cases per center could be sufficient to reach the planned sample size. Were there hospitals with more than 50 cases? Others with less? For sure, there could be centers treating more than 50 cases and others less than 50, however, all the Centers are representative of the Italian situation. Eventually, add this information to the supplementary document. Thank you, no need in our opinion to add information in the supplementary document.

Line 78: correct typo (space after parenthesis): “characteristics ( age, menopausal status, comorbidities);” Done, thank you

If the number of patients screened was 1,071, why do they mention 1,500 in the text? In my opinion, 1,071 remains a significant sample. Thank you, we also believed that at least 1000 patients were sufficient to describe the primary end point. Finally, the total number enrolled was 1071. We modified into the text accordingly

Describe what the F-ratio is about, which will be mentioned later. How to interpret the results? Thank you, detailed in the section “statistical analysis”

Reviewer 2 Report

Comments and Suggestions for Authors

In this manuscript, Cazzaniga et al. conducted an investigation based on GIM-13-AMBRA study to show clinical outcomes among  HER2-negative metastatic breast cancer patients in Italy. The research content of this manuscript is complete and evidence-based. However, further clarification is needed regarding the following matters:

*Tables necessitate further adjustments in certain aspects.

*Table 4, “141%)” should be “141(%)”

*Table 4, why the amount of TNBC is 141 instead of 145?

*In line 22, “HER2-ve” means?

*Figure size should be adjusted.

*Introduction and discussion part should be detailed, e.g. PMID: 37348019 and PMID: 32803712.

Comments on the Quality of English Language

Minor editing of English language required.

Author Response

In this manuscript, Cazzaniga et al. conducted an investigation based on GIM-13-AMBRA study to show clinical outcomes among  HER2-negative metastatic breast cancer patients in Italy. The research content of this manuscript is complete and evidence-based. However, further clarification is needed regarding the following matters:

*Tables necessitate further adjustments in certain aspects.

*Table 4, “141%)” should be “141(%)”, done

*Table 4, why the amount of TNBC is 141 instead of 145? Data regarding 4 TNBC patients are missin in terms of CHT details

*In line 22, “HER2-ve” means? Modified in HER2-, according to the whole text

*Figure size should be adjusted. Done

*Introduction and discussion part should be detailed, e.g. PMID: 37348019 and PMID: 32803712. We are sorry, but the papers you suggested to take inspiration for the introduction are on topics deeply different from our paper. We tried to make simpler our introduction section. Thank you.

Reviewer 3 Report

Comments and Suggestions for Authors

This retrospective study is an interesting approach to previous treatment strategies and could be the basis for future trials in the same network. 

Author Response

REVIEWER 3

This retrospective study is an interesting approach to previous treatment strategies and could be the basis for future trials in the same network

Thank you